# Falsifying computational models of endothelial cell network formation through quantitative comparison with *in vitro* models

Tessa M. Vergroesen[1], Vincent Vermeulen[1], Roeland M. H. Merks[1,2]*

**1** Institute of Biology Leiden, Leiden University, Leiden, The Netherlands, **2** Mathematical Institute, Leiden University, Leiden, The Netherlands

* merksrmh@math.leidenuniv.nl

## Abstract

During angiogenesis, endothelial cells expand the vasculature by migrating from existing blood vessels, proliferating and collectively organizing into new capillaries. *In vitro* and *in vivo* experimentation is instrumental for identifying the molecular players and cell behaviour that regulate angiogenesis. Alongside experimental work, computational and mathematical models of endothelial cell network formation have helped to analyse if the current molecular and cellular understanding of endothelial cell behaviour is sufficient to explain the formation of endothelial cell networks. As input, the models take (a subset of) the current knowledge or hypotheses of single cell behaviour and capture it into a dynamical, mathematical description. As output, they predict the multicellular behaviour following from the actions of many individual cells, *i.e.*, formation of a vascular-like network. Paradoxically, computational modelling based on different assumptions, *i.e.*, completely different, sometimes non-intersecting sets of observed single cell behaviour, can reproduce the same angiogenesis-like multicellular behaviour, making it practically impossible to decide which, if any, of these models is correct. Here we present dynamical analyses of time-lapses of *in vitro* endothelial cell network formation experiments and compare these with dynamic analyses of three mathematical models: (1) the cell elongation model; (2) the contact-inhibited chemotaxis model; and (3) the mechanical cell-cell communication model. We extract a variety of dynamical characteristics of endothelial cell network formation using a custom time-lapse video analysis pipeline in ImageJ. We compare the dynamical network characteristics of the *in vitro* experiments to those of the cellular networks produced by the computational models. We test the response of the *in silico* dynamical cell network characteristics to changes in cell density and make related changes in the *in vitro* experiments. Of the three computational models that we have considered, the cell elongation model best captures the remodelling phase of *in vitro* endothelial cell network formation. Furthermore, in the *in vitro* model, the final size and number of lacunae in the network are independent of the initial cell density. This observation is

**Data availability statement:** The code used for analyzing the experiments is available on a GitHub repository at https://github.com/TMVergroesen/NetworkAnalysis. The code to generate the simulation data is available on a GitHub repository at https://github.com/math-bioleiden/Tissue-Simulation-Toolkit and https://github.com/rmerks/CPM-FEM. The repository of the time-lapses and simulation data is available at Zenodo at https://dx.doi.org/10.5281/zenodo.14809145.

**Funding:** This publication is part of the 'Mathematics-based strategies for repairing tumour blood vessels' project (TV, RM) with project number 865.17.004 to RM of the Vici research program which is financed by the Dutch Research Council (https://www.nwo.nl/). The funders had no role in study design, data collection and analysis, decision to publish, or preparation of the manuscript.

**Competing interests:** I have read the journal's policy and the authors of this manuscript have the following competing interests. TV and VV have declared that no competing interests exist. RM is academic editor of PLOS Computational Biology. These competing interests will not alter adherence to PLOS policies on sharing data and materials.

also reproduced in the cell elongation model, but not in the other two models that we have considered. Altogether, we present an approach to model validation based on comparisons of time-resolved data and variations of model conditions.

## Author Summary

Understanding how blood vessels grow and organise into well-structured networks is crucial for many clinical applications, from wound healing to cancer treatment. The growth of blood vessels, known as angiogenesis, involves endothelial cells migrating, proliferating, and forming new vascular structures. Angiogenesis is studied through biological experiments, but mathematical models are also used to test whether our understanding of cell behaviour is sufficient to explain cell network formation. Models assuming chemotaxis, elongated cell shape, contact inhibition or mechanical cell-cell communication often produce similar results despite the large differences in the underlying assumptions, making it difficult to determine which model is most accurate. In this study, we analysed time-lapse videos of endothelial cell network formation and compared these with three computational models: (1) the cell elongation model, (2) the contact-inhibited chemotaxis model, and (3) the mechanical cell-cell communication model. By examining the dynamics of network formation and testing how they change with cell density, we found that the cell elongation model best captures key aspects of the real-life cell network remodelling process. This approach highlights the importance of time-resolved data in evaluating computational models and provides a framework for refining our understanding of angiogenesis.

## Introduction

During embryogenesis, endothelial precursor cells, angioblasts, migrate and differentiate to form the first blood vessels. This process of blood vessel formation by *de novo* endothelial cell (EC) production, is called vasculogenesis [1]. After vasculogenesis, the primary vascular network is expanded by angiogenesis. Angiogenesis, the formation of new blood vessels from existing ones, is crucial for physiological and pathological mechanisms such as embryonic development, wound healing and tumour growth. Pro-angiogenic factors, including vascular endothelial growth factor (VEGF)-A, stimulate sprouting in ECs, which make up the inner lining of blood vessels [2–4]. The ECs proliferate and migrate into the extracellular environment and form a vascular network that supplies the surrounding tissue with nutrients and oxygen and removes waste products. A key regulatory factor determining the final structure of the new blood vessel network is the composition of the extracellular matrix (ECM), the scaffold of proteins that surrounds the blood vessels and provides structural support to the newly formed multicellular network [5]. Specific

integrin binding sites on ECM proteins like fibronectin and laminin allow cells to exert stresses on the ECM and sense the mechanical properties of their environment [6]. Cells also secrete enzymes which digest the ECM, releasing ECM-retained growth factors, including VEGF-A, that stimulate EC migration [7]. Additionally, cell-cell interactions, like Notch signalling and vascular endothelial cadherin (VE-cadherin) mediated adhesion, regulate individual cell behaviour within the network and separate ECs into leading tip- and proliferating stalk cells [4]. In response to VEGF-A, cells upregulate Delta-like 4 (DLL4), a transmembrane Notch ligand that activates NOTCH1 in neighbouring cells [8]. NOTCH1 activation subsequently downregulates VEGF receptor (VEGFR)-2 expression, reducing the neighbouring cell's susceptibility to VEGF-A, resulting in less angiogenic potential of the cells [8]. Similarly, VE-cadherin binding inhibits VEGFR-2 signalling in neighbouring cells [9]. The combination of extracellular, intercellular and intracellular regulation makes angiogenesis a complex and multiscale process. An understanding of how each of these regulatory mechanisms contribute to the final network formation is necessary for biomedical applications where angiogenesis needs to be stimulated, steered or reduced, like tissue engineering [10] or vascular renormalization [11].

One of the more commonly used *in vitro* models to test the effect of environmental changes on EC network formation is the tube formation assay (TFA). In this assay ECs are placed on top of a basement membrane-like substrate, where the ECs, under the right conditions, spontaneously organise into networks of interconnected tubes [12]. With this assay the influence of various chemical and mechanical stimuli on EC networks has been investigated by comparing the branch density of the final EC network between conditions [13]. In the analysis of TFA outcomes it is often assumed that a higher density of branches indicates a higher angiogenic potential of the cells (*i.e.*, more sprouting). However, the temporal development of the network over time may contain key information on the dynamic processes regulating angiogenesis. Merks *et al.* (2006), therefore, measured the density of branch points in developing vascular networks *in vitro* at regular intervals and observed that the network density dropped slowly over time [14]. Parsa *et al.* (2011), analysed time-lapse videos of *in vitro* EC network formation in more detail, and described the development of EC networks as a carefully orchestrated sequence of five subevents: (1) rearrangement and aggregation, (2) spreading, (3) elongation and formation of cell–cell contacts, (4) plexus stabilization, and (5) plexus reorganization [15]. The first two stages describe the attachment of the cells to the substrate directly after cell seeding. The rate and the extent of cell spreading on the substrate have been linked to substrate stiffness and composition [6], and are assumed to be influenced by the traction forces cells are able to exert on the substrate [16]. Later stages in the tube formation assay rely on a combination of cell-cell and cell-ECM interactions: Rüdiger *et al.* (2020) observed a collapse of EC networks on soft, laminin matrices, confirming the importance of the ECM as a scaffold to provide support for the network to stabilise [5].

Computational modelling is a helpful tool for proposing hypotheses and testing whether (a combination of) proposed developmental mechanisms are sufficient to explain biological observations [17]. Different computational models of angiogenesis all consider angiogenesis as the assembly of ECs into a vascular-like network structure [18–20]. For example, in Manoussaki *et al.* (1996), the *in vitro* matrix remodelling behaviour of ECs on a compliant substrate is described with a continuum model, in which ECs are described as local densities using a system of differential equations [21]. Similarly, in Serini *et al.* (2003), ECs are modelled to migrate upwards a chemoattractant gradient with a continuum model [22]. In Palachanis *et al.* (2015), ECs are considered as identical 2D ellipses, using a single-particle model [23]. Köhn-Luque *et al.*, (2013) used a hybrid continuum and cell-based model to model the effect of VEGF retention in the ECM on EC network formation [7]. In cell-based models of angiogenesis, ECs are described as discrete areas of variable shapes, where multicellular behaviour arises from the response of single cells to inputs from their microenvironment [24,25]. Similarly, Vega *et al.* (2020) used a hybrid continuum and cell-based model to model the interaction between Notch- and VEGF signalling and cell-ECM adhesion [26].

Interestingly, both continuum models, particle-based models, as well as cell-based models are successful in mimicking EC network formation, and different hypotheses, or model inputs, can in some cases lead to a qualitatively similar output. In this work we focus on three cell-based models as previously developed in our group [14,27,28]. The "cell elongation

model" assumes that ECs are attracted towards one another through autocrine/paracrine signalling by secreting a chemoattractant [14]. In this model it was observed that cells self-organise into vascular-like structures if the cells have an elongated shape. Observations by Parsa et al. (2011) support the necessity for ECs to elongate to form networks, as cells elongate before they start forming a network and continue to elongate as the network develops [15]. The "contact inhibition model" also assumes that ECs are attracted to one another through a secreted chemoattractant, but here it was assumed that VE-cadherin binding mediates contact inhibition of chemotaxis, following observations by Dejana et al. (2004) that VE-cadherin-VEGFR2 complex forming limits cell proliferation and promotes quiescence [29] and that in the yolk sac of VE-cadherin double-knockout mice and in in vitro mouse allantois explants ECs failed to form vascular networks and aggregated into isolated islands [28,30]. In disagreement with VEGF guided cell-cell attraction-based models, Rüdiger et al. (2020) observed persistent network formation in the absence of a VEGF-A gradient and they argue that the in vitro network formation is driven by mechanical communication [5]. The importance of cell-ECM interactions is further supported by observations by Stéphanou et al. (2007), where they observed faster and more lacunae in network formation on fibrinogen gels of intermediate rigidity [31]. In light of these observations, the "mechanical cell-cell communication model" assumes that ECs are attracted to one another solely through mechanical interaction with the ECM [27]. In this cell-ECM interaction model ECs induce strain in the ECM through contractile forces and migrate up the strain gradient. Thus, a variety of models can successfully reproduce multicellular network formation starting from different assumptions, all inspired by observed EC behaviour. Therefore, we need to critically examine these models to see which of these assumptions, if any, are correct.

Thus, for a more complete understanding of the integration of different mechanisms driving EC network formation, we must quantitatively compare these computational models to in vitro experiments by systematically changing model parameters and in vitro experimental setup, and by iteratively refining the models based on the outcomes of the comparison. As an initial step, we will critically examine three models originating from our own research group: (1) The cell elongation model [14]; (2) the contact inhibition model [28]; and (3) the mechanical cell-cell communication model [27]. We select model parameters from the original work [14,27,28] and from published global sensitivity analyses [32]. Then we fit the models to our own in vitro observations, and finally we present additional parameter examinations. For the comparison, dynamical analyses of in vitro experiments are required. Here we present new dynamic time-lapse videos of tube formation assays and a new analysis pipeline based on ImageJ, and demonstrate its use for model selection for network formation. Commercial and open-source image analysis tools are available for quantitative analyses of EC networks [33–36]. However, these tools are designed for single frame end-point analyses and therefore less suitable for high-throughput dynamical analyses of large quantities of data. Our novel image analysis pipeline is suitable to dynamically analyse and compare in vitro and simulated networks. The pipeline allows us to reliably extract network features over time, such as the number of branches and their length and the average size and number of lacunae. In this study we use this pipeline to compare the output of three cellular Potts models of 2D EC network formation [14,27,28] to in vitro experiments over time at different cell densities and find that the distance at which cells are able to communicate with each other determines the features of the final, stabilised network, but the speed and manner at which the networks stabilise depends on other characteristics of the cell communication, like the speed at which the cell attracting signal spreads.

## Results

### Quantification of in vitro endothelial cell network dynamics

To understand and compare the dynamic behaviour during in vitro and in silico EC network formation, we developed an image analysis pipeline to extract network characteristics from 2D EC network formation time-lapses and applied it to in vitro and in silico videos. To capture in vitro EC network formation, ECs were seeded on a Matrigel matrix and imaged in phase contrast for 24 hours with an interval of 15 minutes (Fig 1A). The ECs in the time-lapse images were segmented from the background using a combination of Gaussian denoising and local variance extraction, followed

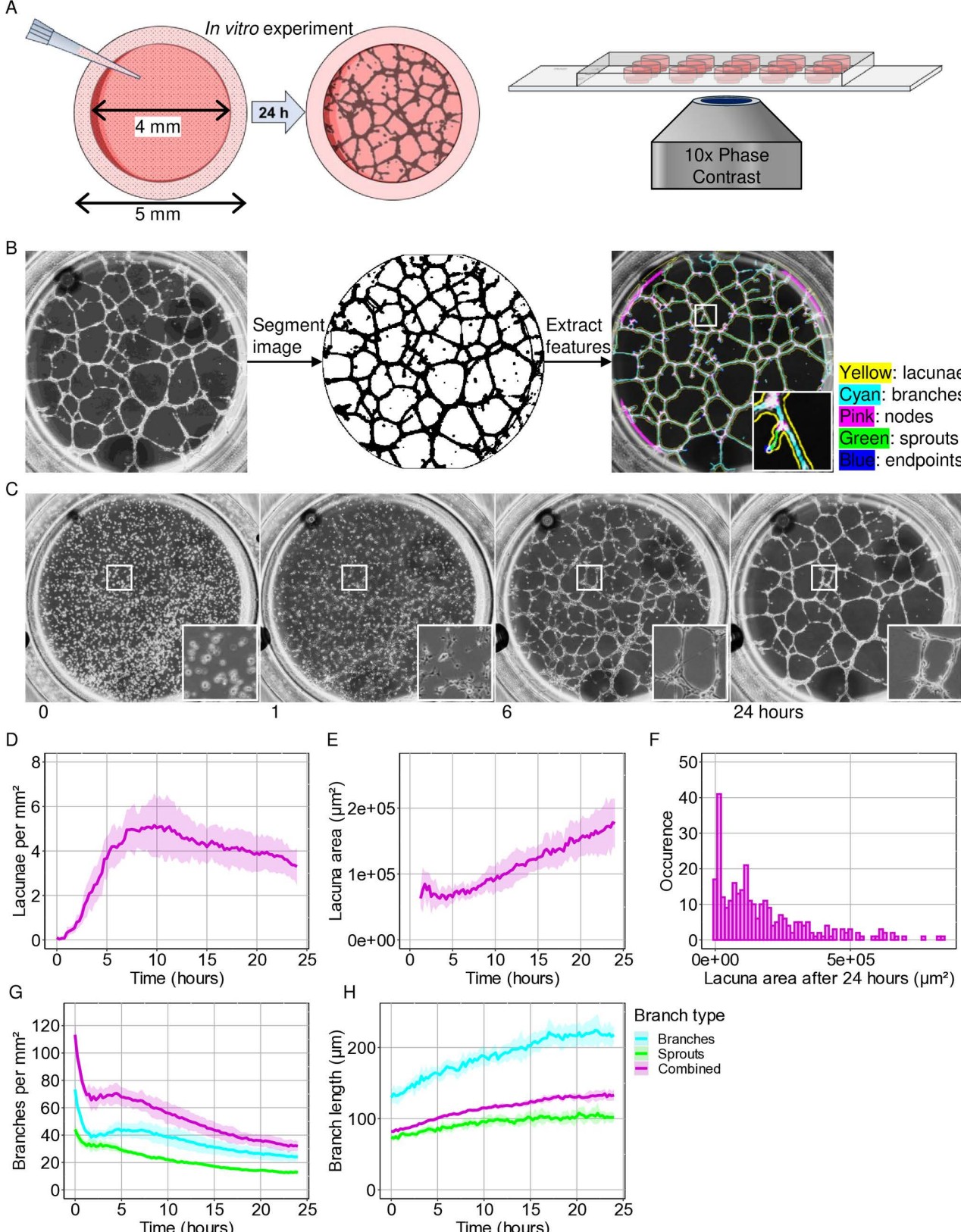

**Fig 1. Image analysis pipeline for *in vitro* phase contrast time-lapses. A)** Graphic summary of *in vitro* experiment. **B)** Networks were segmented from the background and labelled in FIJI. Insert is 350 x 350 μm². **C)** ECs were seeded on Matrigel, and network formation is captured

in phase contrast for 24 hours with intervals of 15 minutes. Insert is 500 x 500 µm². D-G) Quantitative analysis of (D-F) lacunae and (G-H) branches under standard conditions. Shaded area represents a standard deviation of eight wells on one slide.

by Huang thresholding (S1A Fig). Network features were extracted from the segmented time-lapses for each time step using FIJI plugins Analyze Skeleton [37] and Analyze Particles (Fig 1B and S1 Video). For the *in silico* networks, we selected three cell-based computational models of 2D EC network formation [14,27,28] and extracted the network characteristics directly from the output images using the same image analysis pipeline as the segmented *in vitro* time-lapses.

In the *in vitro* culture, under standard conditions, the ECs started forming cell-cell connections within the first hour after seeding on a Matrigel matrix and thereafter self-organised into fully connected networks within six hours (Fig 1C). We were able to divide the development of the network characteristics over time in three distinct parts, which we describe following Parsa *et al.* (2011) [15], as: (1) The first hour resembled stages 1 and 2, "rearrangement and aggregation", followed by "spreading" (Fig 1C). In these stages the ECs were randomly dispersed on top of the Matrigel and started to form attachments to the ECM. (2) The next 6 hours matched Parsa's stage 3, "elongation and formation of cell-cell contact". During this stage new connections formed between ECs, which is reflected in an increase in the number of lacunae from $0.03 \pm 0.09$ to $5.16 \pm 1.05$ per mm² (Fig 1D). (3) Hereafter, the number of lacunae decreased linearly in time ($R^2 = 0.82$, p-value < 0.001), resembling Parsa's stage 4 and 5: "plexus stabilization" and "reorganization". The average lacuna area displayed a linear increase due to a higher occurrence of lacuna merging and smaller lacunae closing compared to the formation of new lacunae (Fig 1E) ($R^2 = 0.89$, p-value < 0.001). The number of branches within the network continuously decreased and their average length increased (Fig 1G and 1H). During the first three stages branches predominantly increased in length due to merging of branch sprouts, but during stages 4 and 5 we observed an increase in merging of existing branches.

### *In silico* models can be adjusted to resemble *in vitro* networks

To compare the dynamics of *in silico* networks to the *in vitro* networks we selected model parameters that mimic the *in vitro* experiment. To represent the inner well of the *in vitro* experiments (Fig 1A), ECs were simulated within a circular field with a diameter of 1900 lattice sites for the two chemotaxis models (1 lattice site = 2 x 2 µm) and 760 lattice sites for the mechanical model (1 lattice site = 5 x 5 µm) (Fig 2A and S2-4 Videos). To determine the target area and length of the simulated cells, we measured cell areas and diameters in fluorescently labelled ECs within network time-lapses (S2A Fig). We observed an average cell area of $1052 \pm 596$ µm² after three hours (S2B Fig), with a cell length of $90 \pm 21$ µm for elongated cells (S2C Fig). Based on these measurements we set the target area for the simulated cells to 250 lattice sites (chemotaxis models) and 40 lattice sites (mechanical model), which corresponds to a physical area of 1000 µm². In the cell elongation model, we set the target length to 45 lattice sites, which corresponds to 90 µm.

In the cell elongation model, ECs elongated towards a target length, secreted a chemoattractant and migrated upwards the chemoattractant gradient [14]. In the contact inhibition model ECs secreted a chemoattractant and migrated upwards the chemoattractant gradient, but locally inhibited chemotaxis in their neighbours (Fig 2A) [28]. To find the parameters that describe the chemoattractant in the two chemotaxis models for which the models most resemble the *in vitro* networks (*i.e.*, the diffusion coefficient, the decay rate and the secretion rate) we investigated the effect of the diffusion length of the chemoattractant on the network formation. Diffusion length ($I_D$) depends on the diffusion coefficient (D) and the decay rate ($\in$) of the chemoattractant according to:

$$I_D = \sqrt{\frac{D}{\in}}$$

(1)

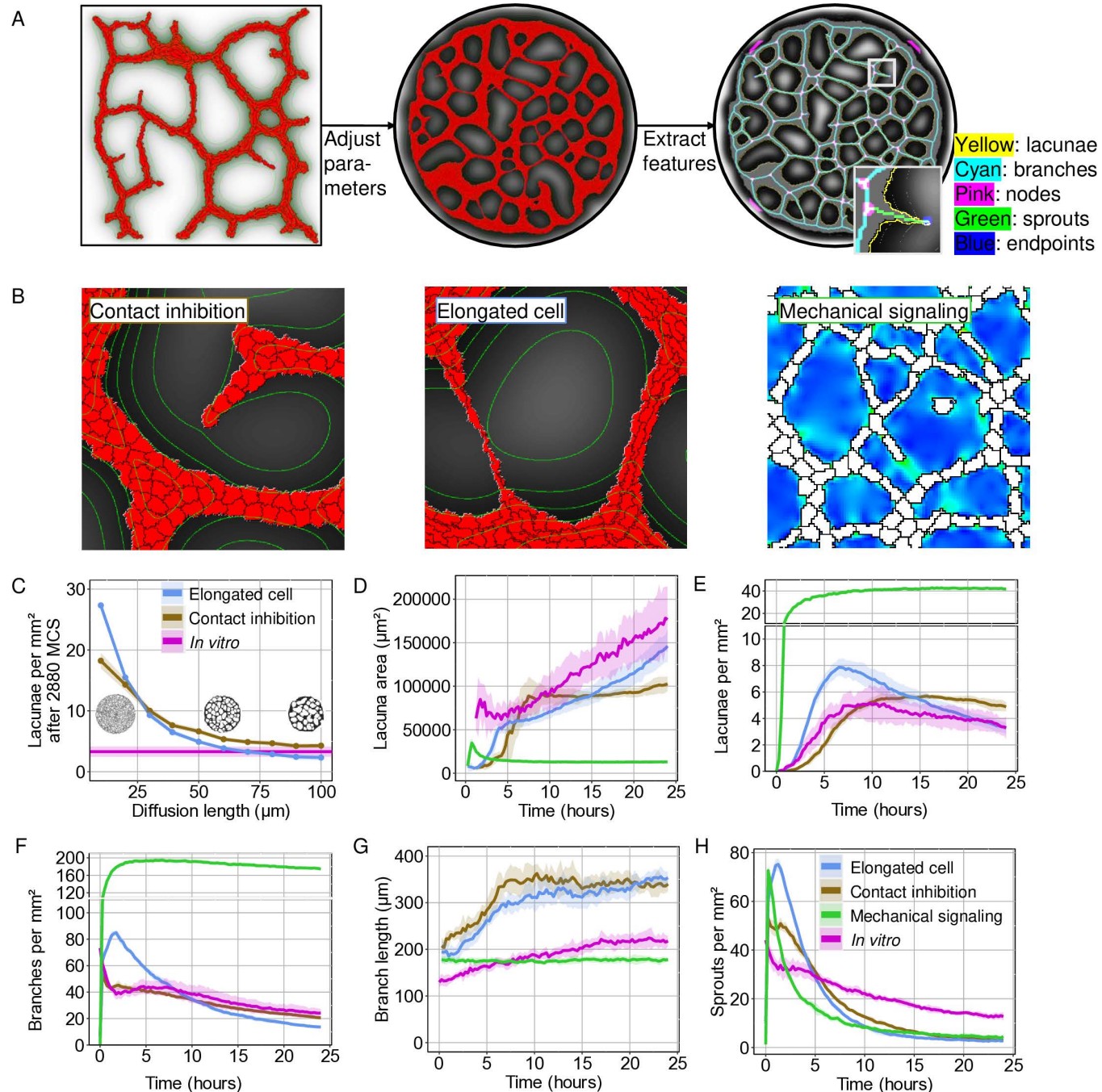

**Fig 2. Quantitative comparisons between two chemotaxis-based models and a mechanical model of 2D EC network formation to *in vitro*
time-lapse results.** A) The model parameters are adjusted to resemble the *in vitro* situation and network characteristics are extracted using the same
image analysis pipeline as the *in vitro* networks. B) Simulated ECs were placed on a circular grid and run for 2880 Monte Carlo steps (MCS). In the
chemotaxis models ECs (red) secrete a chemoattractant (grey) that diffuses through the medium (white). In the mechanical model ECs (white) exert
strain on their environment (green/blue heatmap) (Field area is 500 x 500 µm²). C) A comparison of the number of lacunae per mm² for chemoattractant
diffusion lengths ranging from 10 µm to 100 µm for the cell elongation model and the contact inhibition model after 2880 MCS. In magenta the *in vitro*
number of lacunae per mm² after 24 hours. Inserts show the cell elongation model for a diffusion length of 10, 50 and 100 µm. D-G) Quantitative analysis
of (D, E) lacunae and (F, G) branches *in vitro* (magenta) and in the cell elongation model (blue), the contact inhibition model (golden) and the mechanical
model (green). H) The number of sprouts, branches connected to a single node *in vitro*, in the cell elongation model, the contact inhibition model and the

mechanical model. ($D = 5.0 \cdot 10^{-13} m^2 s^{-1}$; $\in = 1.02 \cdot 10^{-4} s^{-1}$; $\alpha = 1 \cdot 10^{-3} s^{-1}$). Shaded area represents a standard deviation of eight simulations or eight *in vitro* time-lapses.

Therefore, we altered the diffusion length from $l_D = 10 \mu m$ to $l_D = 100 \mu m$ with steps of $10 \mu m$ by varying the decay rate of the chemoattractant, while keeping the diffusion coefficient constant at $D = 5 \cdot 10^{-13} m^2 s^{-1}$. (Fig 2C). For a full list of the selected parameters, see S1 Table. *In vitro*, the average number of lacunae per mm² after 24 hours was 3.30 ± 0.83, with an average lacuna area of 0.18 ± 0.04 mm² (Fig 1D and 1E). When we increased the diffusion length in the two chemotaxis model, we observed that the average number of lacunae decreased (Fig 2C), in agreement with previous work [14,28,32].

Based on the outcome of the comparison of the number of lacunae (Fig 2C) and branches (S2D Fig) per mm², we selected a decay rate of $\in = 1.02 \cdot 10^{-4} s^{-1}$, which corresponds to a diffusion length of $l_D = 70$ μm. This resulted in an average number of lacunae per mm² after 2880 MCS of $3.27 \pm 0.47$, with an average area of $0.15 \pm 0.02$ mm² for the cell elongation model and $4.88 \pm 0.35$ lacunae per mm², with an average area of $0.10 \pm 0.01$ mm² for the contact inhibition model (Fig 2D and 2E). With the selected diffusion length of $l_D = 70$ μm, we observed some similarities between the dynamics of the model outcomes and the *in vitro* networks: During the initial relaxation time (100 MCS) the ECs in the models reached their target size and shape, which mimics stages 2 and 3 of the 5 stages described in [15]: cell spreading and elongation. Next, we looked at the formation of cell-cell connections as ECs started excreting a chemoattractant. The increase in cell-cell connections is reflected by a decrease in number of branches (Fig 2F). The transition towards stage 4, stabilization of the network, was more gradual in the contact inhibition model than in the *in vitro* networks and in the cell elongation model (Fig 2E). The number of lacunae decreased faster in the cell elongation model than in the contact inhibition model (Fig 2E) and for the first 1680 MCS we observed fewer and larger lacunae in the contact inhibition model than in the cell elongation model. The number of branches was initially larger in the cell elongation model than in the contact inhibition model. In agreement with the *in vitro* networks, the branch number decreased quasi-exponentially in the cell elongation model, whereas in the contact inhibition model the number of branches decreased more linearly (Fig 2F). Both chemotaxis-based models have longer branches than the *in vitro* networks (Fig 2G). This difference in number of branches and their length could partially be explained by a higher density of sprouts detected *in vitro* than in the models (Fig 2H). Sprouts are branches that are connected to a single node and the Analyze Skeleton plugin splits an existing branch into two where a sprout appears, resulting in two shorter branches.

In the mechanical model, the ECs are assumed to stiffen the ECM through the exertion of traction forces on their environment and move upwards the stiffness gradient (Fig 2B) [27]. We selected a Young's modulus of 12 kPa, with a Poisson's ratio of 0.45, based on the original work. For a full list of the selected parameters, see S2 Table. For the selected parameters, cells formed a dense network of branches with small lacunae. The composition of this network was similar to an early-stage EC network. This suggests that in the present form of the mechanical model, the mechanical model successfully reproduces the network formation phases (phases 1-3), but it did not describe the remodelling phase well.

Another noticeable difference between the three cellular Potts models and the *in vitro* EC networks was the composition and thickness of the branches (Figs 1B and 2B, and S3). *In vitro* cells are able to move on top of each other to form a semi-3D structure, consisting of thin branches and thick nodes. This is reflected in a decrease in the cell-covered area over time (S2E Fig). In the strictly 2D *in silico* models, however, cells are unable to move on top of each other and the cell-covered area remains constant (S2E Fig). Therefore, the models form increasingly thick branches and larger nodes as the cell density increases (S3 Fig).

## Increasing cell density results in an initial increase in number of branches and lacunae *in vitro*

To investigate whether the models respond similarly to a change in the initial conditions as *in vitro* networks, we next analysed network formation at different cell densities. Previous *in vitro* studies on primary ECs observed an inability for

the cells to form tubes under or over a threshold density, similar to a percolation threshold [5,22]. Consistent with these previous results, we observed that HMEC-1 cells retained the ability to form cell-cell connections and short branches at 1000 ECs per well (80 cells/mm²) (Fig 3A). However, they were not able to form a network spanning the full well. At a cell density of more than 2000 ECs per well (160 cells mm²), we observed the development of a fully connected network spanning the entire well (Fig 3A). With an increase in cells density a higher number of lacunae and branches formed within the first hours (Fig 3B and 3D). However, for over 4000 cells per well (320 cells mm²), the networks remodelled to the same number of lacunae and branches and average lacuna and branch sizes (Fig 3B-3D).

### The cell elongation model resembles the *in vitro* networks in its response to changes in cell density

Whereas a higher cell density caused an initial increase in the number of branches per mm² for all three *in silico* models, this difference diminished as the network remodelled in the two chemotaxis-based models at a diffusion length of 70 μm (Fig 4B and 4E), similar to what we observed *in vitro* (Fig 3B). The mechanical model had a much higher number of branches per mm², and the difference between branches per mm² for different cell densities converged much slower than in the chemotaxis-based models and the *in vitro* time-lapses (Figs 3B and 4A–4I). Out of the three models, the later dynamics of the cell elongation model most resembled the *in vitro* network remodelling, which appears to be independent to a change in cell density, even when the model is subjected to a range of diffusion lengths (Figs 4A–4C, and S4).

## Discussion

In biomedical applications, like tissue engineering [10] or tumour vascular normalization [11], angiogenesis is stimulated, redirected or reduced to properly vascularise the tissue of interest. Computational models can help understand how different environmental factors and cellular cues affect various aspects of EC migration and network formation, and help

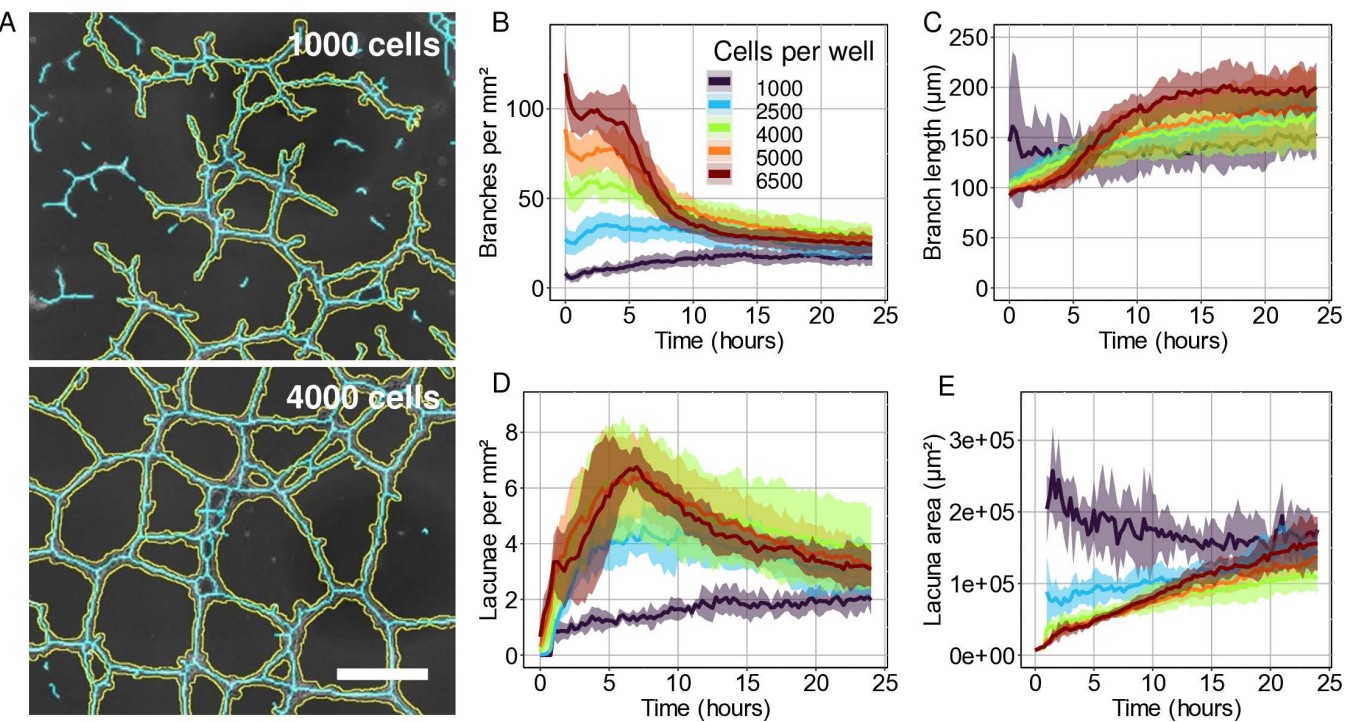

**Fig 3. A comparison of the response of *in vitro* networks and model outcomes to higher EC seeding density.** A) Segmented *in vitro* networks with an inner well cell count of 1000 (top) and 4000 (bottom). B-E) Quantitative analysis of (B, C) branches and (D, E) lacunae for different cell densities. Scale bar = 500 μm. Shaded area represents standard deviation of 5 or 6 wells from two separate experiments.

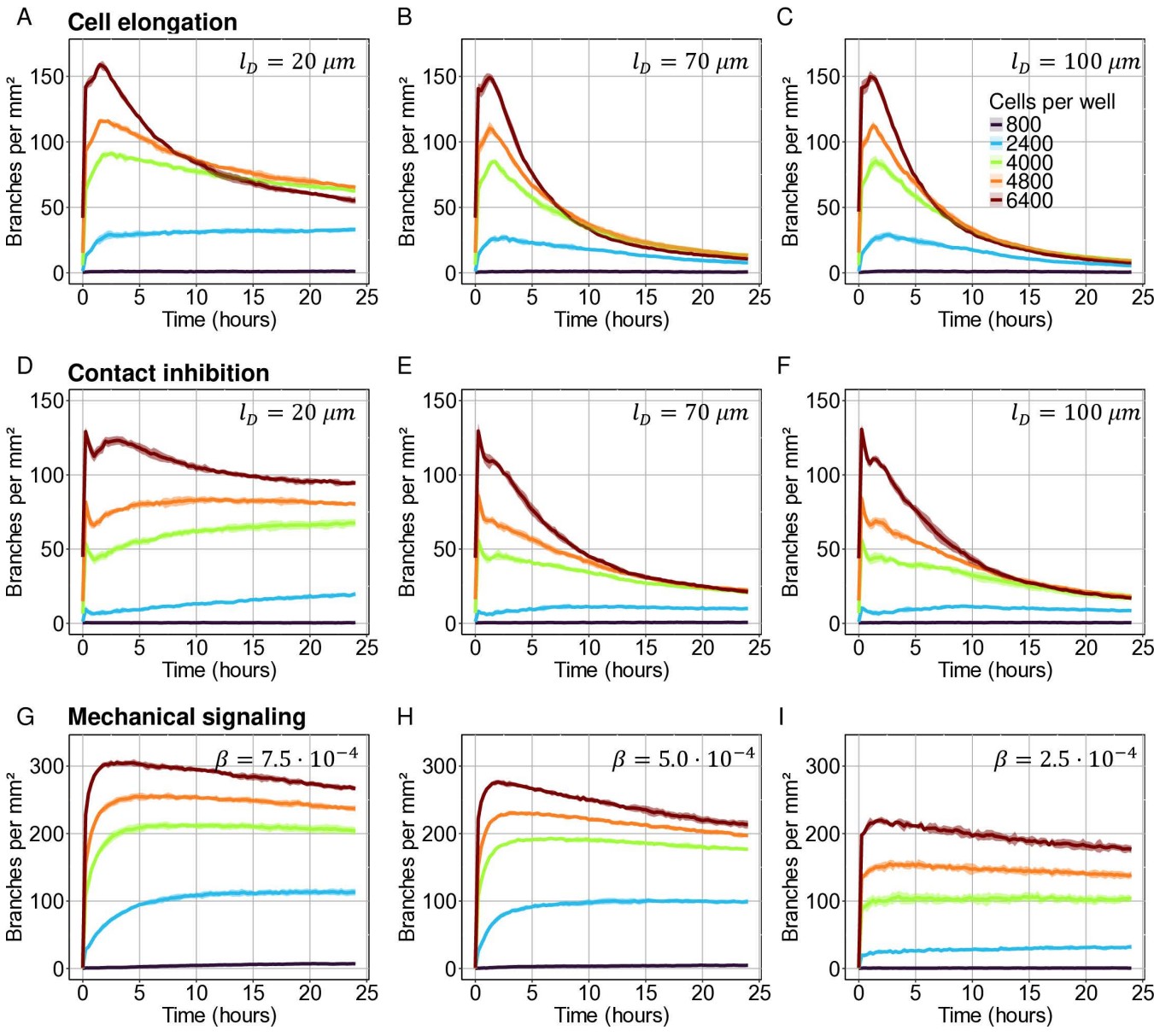

**Fig 4. A comparison of the response of model outcomes to a higher EC density.** Quantitative analysis of the number of branches for different cell densities with different diffusion lengths ($l_D$) in the (A-C) cell elongation model and (D-F) the contact inhibition model, and (G-I) different stiffness sensitivity (β) in the mechanical model. $D = 5.0 \cdot 10^{-13} m^2 s^{-1}; \epsilon = 1.02 \cdot 10^{-4} s^{-1}; \alpha = 1 \cdot 10^{-3} s^{-1}$. Shaded area represents the standard deviation of six simulations.

identify effective regulatory targets. Many computational models of EC network formation, based on assumptions like cell-cell attraction [28], cell shape effects [14,23], cell-ECM interaction [21,27], or a combination of the former [38], have been successful at simulating the formation of EC networks that qualitatively resemble those observed *in vitro*. To obtain more insight into the extent to which the mechanisms described by these computational models are a sufficient explanation for EC network formation *in vitro* we selected three previously published, cell-based models of 2D EC network formation and compared these with time-lapse data of EC network formation *in vitro*: (1) the cell elongation model [14]; (2) the

contact-inhibited chemotaxis model [28]; and (3) the mechanical cell-cell communication model [27]. We examined their dynamical behaviour using a range of quantifications and compared it to time-lapses data of *in vitro* EC network formation.

To quantify dynamical features of both *in vitro* EC network formation and outcomes of cellular Potts models of 2D EC network formation, we developed a custom image analysis pipeline (Figs 1B and 2A, and S1A). To compare the models to our *in vitro* time-lapses we selected parameters from the original work [14,27,28] and from published sensitivity analyses [39], and fitted parameters to our own *in vitro* observations (S2A-S2C Fig). We used the outcome of our initial dynamical analysis of the *in vitro* and simulated EC networks to find a parameter space in which the network features of the chemotaxis-based model outcomes resembled those of the *in vitro* EC networks (Figs 2C and S2D). For the mechanical cell-cell communication model we were not able to find such a parameter regime. The mechanical model was very sensitive and often lost its ability to form networks after small changes in the explored parameters. Its high computational cost also made it difficult to perform large scale parameter sweeps.

We then asked how the *in vitro* model responded to systematic changes in the culture conditions, with the aim of mimicking the same conditions *in silico*. Among the conditions tested (S5 Fig) only cell seeding density affected network formation *in silico* and *in vitro* to a sufficient degree to allow for assessment of the correctness of the models (Figs 3 and 4). Interestingly, both chemotaxis-based models showed similar dynamic behaviour as the *in vitro* networks for the selected parameter regime (Figs 3 and 4, and S4). They showed an initial cell-cell contact forming phase, followed by a remodelling phase, similar to Parsa's stages 3 to 5 of *in vitro* network formation: "elongation and formation of cell-cell contacts", "plexus stabilization", and "plexus reorganization". In the mechanical model, however, the ECs quickly form connections, mimicking early *in vitro* network formation, but there is little network remodelling in the model outcomes.

We found that, among the models tested, the dynamic behaviour of the lacunae in the cell elongation model matched the dynamics of the *in vitro* networks most closely (Fig 2D and 2E). In the cell elongation model [14] the ECs actively elongate before they migrate upwards a chemoattractant gradient. Because of their shape, the elongated ECs can form long-range connections to other ECs and can thus form branches and close lacunae more rapidly than rounded ECs (Fig 2E and 2F). Due to the slower movement along the short axis for elongated cells, branches are not very mobile and merge slowly as the network stabilises [40]. This results in a gradual decrease in lacunae per mm², which mimics the remodelling phase of *in vitro* network formation quite well (Fig 2E). In the contact inhibition model [28] however, network remodelling proceeds more slowly than in the elongated EC model (Fig 2D and 2E). Also, in the contact inhibition model, for a short diffusion length ($l_D = 20$ µm), the number of branches per mm² remains dependent on the cell density (Fig 4D), which is not observed *in vitro* or in the cell elongation model. The cell elongation model also predicts that networks can form at lower cell densities than the contact inhibition model (Figs 4 and S4). Thus, the cell elongation seems more robust in reproducing the remodelling stage of *in vitro* EC network formation than the contact inhibition model.

There were also clear discrepancies between the cell elongation model and *in vitro* network formation. For example, Parsa *et al.* (2011) observed that *in vitro* ECs aggregate and spread first, before elongating [15]. Thus, *in vitro*, the earliest stages of network formation seem not driven by elongation, as assumed by the cell elongation model. Also, the cell elongation model was unable to explain the observation that network formation depends on substrate stiffness [5,41,42], as well as substrate density (S5F Fig) [6] and substrate thickness [43]. Such observations may be better explained by models based on mechanical cell-cell communication [21,27]. Rüdiger *et al.* (2020) argue that mechanical forces are the main driver behind *in vitro* EC network formation [5]. In this study they continuously perfused the ECs during *in vitro* network formation to wash away any possible chemoattractant gradient, and network formation still occurs. To further argue for mechanical regulation, they show that ECs compress the substrate up to a distance of 30 µm, with measurable ECM displacement up to 100 µm. These distances fall within the chemical signalling range of what we observed in the two chemotaxis models similar to the *in vitro* networks (Figs 2C and S2D). In the mechanical model we observed strains up to 40 µm from the cell boundary in the investigated parameter regime (S2F Fig). However, this was not enough to form networks similar to those observed *in vitro* (Fig 2). Finding parameter regimes for the mechanical model in which it

quantitatively reproduces the *in vitro* dynamics might make it possible to explore the effect of ECM straining on EC network formation in more detail. To find new parameter regimes in the mechanical model we could combine our presented image-analysis pipeline with topological data analysis [44] and novel high-throughput parameter inference methods such as FitMultiCell [45], which combines the iteratively changing of parameters, the running of simulations and the assessment of the model outcomes in one tool, to streamline the model fitting process. Also, a number of improvements are possible in the mechanical model, which so far focused on synthetic, fully elastic substrates such as polyacrylamide gels, which cannot be deformed or remodelled. In cell cultures using natural ECM materials such as Matrigel, the ECs tend to deform and remodel the substrate over time [5,46], altering the substrate's material properties and its response to stress. The absence of ECM remodelling in the mechanical model could explain why the EC networks in the mechanical model stabilises relative quickly, whereas *in vitro* and in the two chemotaxis-based models the EC networks continuously evolve as the simulated chemoattractant gradient gradually stabilises over time. The effect of matrix remodelling, in the form of strain-stiffening, on EC network formation dynamics could be investigated *in vitro* with synthetic gels with tuneable stiffness [47,48]. With these gels the travel distance of mechanical perturbations on the system can be altered systematically, to measure its effect on EC network formation [16]. With the use of synthetic gels, it is also possible to wash away any chemoattractant from the system, without the possible interference of ECM retention, which might have influenced the network formation in Rüdiger *et al*. (2020). *In silico* models have considered matrix remodelling by using continuum equations of matrix advection due to pulling by the ECs [21,49]. A similar approach was integrated in a recent hybrid CPM-PDE model of network formation [38]. In these models, the number of junctions was reported to increase over time, in contrast to our *in vitro* observations (Fig 1C-1H) and those of others [15,50], where the network remodels into a less dense network within 24 hours. In our ongoing work, we are integrating detailed models of the structural remodelling of fibrous ECMs into models of angiogenesis [51,52], which in the future may give more insight into the potential role of matrix remodelling.

In the two chemotaxis-based models we assumed that each MCS represents 30 seconds in real time and each pixel is 2 x 2 μm. This was chosen such that the mean speed of the cells became $5\mu m/h^{-1}$, which, it was previously argued, complies with *in vivo* observations [14,53]. This allows us to translate the physical processes like chemoattractant diffusion and decay rate to physical units ($m^2 s^{-1}$; $s^{-1}$). We fixed the diffusion coefficient at $D = 5 \cdot 10^{-13} m^2 s^{-1}$. The main driver behind angiogenesis is generally assumed to be VEGF-A [54]. The diffusion coefficient of VEGF-A is in the order of $10^{-11} m^2 s^{-1}$ in collagen-I, $10^{-10} m^2 s^{-1}$ in Matrigel and $10^{-9} m^2 s^{-1}$ in water [55–57], which is 100 to 10000-fold higher than what we implemented in the model. However, ECM and EC membrane receptor retention of VEGF-A could affect VEGF-A diffusion, resulting in a smaller effective diffusion coefficient and decay rate [7]. Köhn-Luque *et al*., (2013) show an aggregation of fluorescently labelled VEGF in the substrate surrounding HUVECs on Matrigel within 5-10 minutes after VEGF addition [7]. Additionally, it is yet unclear whether VEGF-A diffusion is the main driver behind angiogenic network formation *in vitro*. Especially since Rüdiger *et al*. (2020) observed EC network formation under constant perfusion to avoid the emergence of a chemical gradient [5]. With a diffusion coefficient of $D = 5 \cdot 10^{-13} m^2 s^{-1}$ we observe simulated networks most like *in vitro* at a diffusion length of $l_D = 70 \mu m$ (Fig 2C). With a diffusion coefficient of $D = 10^{-10} m^2 s^{-1}$, the decay rate would need to be $\in = 2 \cdot 10^{-2} s^{-1}$ for a diffusion length of $l_D = 70 \mu m$, since $l_D = \sqrt{\frac{D}{\in}}$, which is much faster than the 90 minute half-life of VEGF-A measured *in vitro* [58]. *In vivo*, VEGF-A has been shown to have a half-life of 33.7 minutes [59].

There are some steps in this image analysis pipeline that require optimization. For example, we implemented a watershed step in the pipeline to correct for breaks in EC branches where thin stretched-out ECs are misinterpreted as background during segmentation. This watershed step can result in an overrepresentation of connections between branches, but overall, we observe an accurate and consistent result (S1B Fig). To circumvent the need for segmentation corrections and other suboptimal parameter settings in the segmentation a neural network could be trained to perform the segmentation steps necessary for the network quantification [60]. This network can be trained to recognise thin stretches of ECs where conventional image analysis methods, which rely mostly on contrast, might mistake them for background.

However, the training of a neural network requires a lot of correctly annotated training data and undertrained networks will give suboptimal results. More interestingly, a neural network could be trained to recognise network features that are more difficult to extract from segmented images automatically, like cell division events. Another interesting addition to the image analysis pipeline would be the inclusion of topological data analysis (TDA) [44]. The time-lapses of the segmented branches and lacunae could be regarded as multidimensional graphs suitable for further analysis using TDA. TDA could also be expanded to work on 3D time-lapses to analyse 3D *in vitro* EC sprouting assays [61,62], 3D vessels-on-a-chip sprouting assays [63] or even *in vivo* systems, like the developing zebrafish embryo [64]. 3D *in vitro* systems have the benefit of both having a high degree of control over the experimental environment, as allowing for 3D vascular growth and EC migration. ECs *in vivo* migrate both in 2D over the basal membrane, as in 3D during sprouting into the ECM [65]. 2D models of EC network formation can be regarded as a projection of a 3D system and adapted to study different migration types, but recent improvements in computational power have allowed the increased usage of more complex 3D models of angiogenesis [66,67]. 3D models, both *in vitro*, *in vivo* and *in silico* allow for a greater understanding of the complex interactions ECs have with the fibrous ECM during sprouting [68].

Conventional tube formation assays to assess the sprouting ability of ECs are usually only examined for a single characteristic (*e.g.* the number of branches) at a single time point [33]. However, we observed that network features converged over time (Fig 3B–3E). This indicates that an endpoint observation might not suffice to examine the influence of a compound or knockout on collected cell behaviour before a stable network has been formed. We also observed variation between different samples of the same experimental conditions. This can be explained by variations in fitness of different passage numbers of ECs [13], handling of the ECs prior to the experiment, and small fluctuations in cell seeding density. Cell density naturally affects the distance between cells right after seeding, and can therefore affect the ability of cells to sense their neighbours at the start of the experiment [5,13,15]. Cells also influence the experimental environment by digesting nutrients and excreting signalling molecules [69], but also by actively remodelling the ECM [5,70]. Therefore, it is important that the exact number of seeded ECs is also considered when comparing samples within 6 hours after seeding or at low cell densities.

In this work we have shown the importance of dynamic analyses of EC network formation by showing how EC network features converge over time. We hope to have paved the way for a more systematic comparison of computational models to biological experiments, and we have indicated areas of improvement for existing cellular Potts models of 2D EC network formation. In the future we plan to expand our analysis of EC networks through more sophisticated data-analysis techniques such as topological data analysis [44] and Sobol' analysis [32] for falsifying and refining our understanding of EC network formation using *in vitro* and *in silico* modelling in hopes that such techniques can be generalised to the dynamics of other models of biological morphogenesis.

## Methods

### Cell culture

Immortalised human microvascular endothelial cells (HMEC-1) were maintained in T75 flasks (Sarstedt Inc) in MCDB-131 medium (Gibco) supplemented with 10% fetal calf serum (FCS), 10 ng/mL epidermal growth factor (EGF) (Sigma-Aldrich), 1 μg/mL hydrocortisone (HC) (Thermo Scientific Chemicals), 1% GlutaMAX (Gibco) and 1% penicillin/ streptomycin (Gibco). Cells were cultured in a humidified incubator at 37 °C and 5% CO2 and split twice per week. Passages used were between 9 and 17.

### Tube formation assay

Cells at 70-80% confluency were plated on growth factor reduced (GFR) Matrigel (Corning) on Angiogenesis μ-slides (Ibidi) at a density of 10,000 cells per well unless stated otherwise. For fluorescent plasma membrane labelling cells were incubated with PKH67, PKH26 or CellVue Claret (Sigma-Aldrich) in Diluent C as described in [15].

## Imaging

All images were captured at a magnification of 10X using a Nikon Ti inverted phase contrast microscope equipped with a DS-Ri2 camera or a Zeiss Axio Observer inverted phase contrast microscope equipped with an AxioCam 705 mono camera with a motorised stage for 24 hours with 15-minute intervals. Fluorescent images were acquired using the Zeiss Colibri multicolour LED light source. Full well images were created by stitching multiple fields together with the NIS-elements and ZEN Blue software.

## Network characterization

Time-lapse images of the full wells were processed and quantified using a custom Fiji/ImageJ [71] pipeline (S1A Fig). For the segmentation of cells from background the following steps were performed for each slice in the time-lapse images: (1) High-frequency features were amplified by first convoluting the phase-contrast image with a Gaussian filter with sigma value of 10 μm, then subtracting the resulting low-frequency image from the original image and finally adding the resulting high-frequency image back to the original image. (2) Hereafter, the local variance in the sharpened images was computed with a radius of 8.75 μm. The local variances were convoluted with a Gaussian filter with a sigma value of 3.5 μm. (3) From the resulting image a threshold was computed with automatic Huang thresholding to create a segmented mask of the network. (4) The mask was then corrected for fused lacunae by producing a distance map of the lacunae larger than 2000 μm² and watershed segmenting the distance map with a threshold set to 60. Hereafter, the smaller lacunae were added back to the watershed-segmented network for the final mask. The watershed step was omitted for the first hour of the time-lapse, since no lacunae have formed yet. The branches and nodes within the binary masks were analysed with the Skeletonize 2D/3D and Analyze Skeleton (2D/3D) [37] plugins applied on a ten times scaled down image to avoid a very noisy skeleton. Lacunae between 4000 μm² and 11.3 mm² with a circularity between 0.2 and 1.0 were analysed with the Analyze Particles plugin. The resulting output includes the number, size, position and shape of the lacunae, and the number, length, position and connectivity of the branches. All results were exported to CSV files for further analysis in RStudio [72]. This analysis pipeline was automated to process large quantities of time-lapse images sequentially without user input to maximise reproducibility and minimise personal bias in the data analysis. For the validation of the pipeline, we had six independent participants manually encircle three lacunae and compared the average lacuna area of their selections to automatically segmented images. The automatic segmentation underestimated the lacunae as selected by the participants by $4.2 \pm 11.7\%$ of the average area (n = 15). The difference between two different participants encircling the same lacunae was $4.5 \pm 2\%$ of average area (n = 3) (S1B Fig).

## Computational model

The three computational models of EC network formation [14,27,28] are all based on hybrid cellular Potts models (CPMs). Hybrid CPMs couple a dynamical description of cell behaviour based on the CPM with a detailed model of the cellular micro-environment, specifically the concentration of extracellular signalling molecules or strains and stresses in the ECM. The cellular Potts model is a cell-based model that focuses on modelling cell-cell interactions and predicting collective cell behaviour [73]. Cells are represented as collections of lattice sites $\vec{x}$ with an identical index $\sigma_x$ on a grid, where each index labels a single cell. Each lattice site represents an area of 2 x 2 μm² in the chemotaxis models and 5 x 5 μm² in the mechanical model. To mimic the circular *in vitro* dishes, we labelled all lattice sites outside of a circle with a diameter of 3800 μm as boundary sites $\sigma_x = -1$ in the cell elongation and contact inhibition models and as a fixed node in the mechanical model.

Apart from the shapes of the simulation domains, models were used as described previously [14,27,28]. Briefly, during a Monte Carlo step (MCS), a time-step in the simulation representing 30 seconds, cells change their shape and move through the lattice by attempting to copy its index to a neighbouring lattice site. There are N copy attempts per MCS, with

N the number of lattice sites. If the copy attempt results in a lower effective energy of the system, the copy attempt is accepted. With an effective energy H:

$$H = \sum_{\vec{x}, \vec{x}'} J_{\sigma_{\vec{x}}, \sigma_{\vec{x}'}} \left(1 - \delta_{\sigma_{\vec{x}}, \sigma_{\vec{x}'}}\right) + \lambda_A \sum_{\sigma} (a_\sigma - A_\sigma)^2 + \lambda_L \sum_{\sigma} (l_\sigma - L_\sigma)^2$$

(2)

where $J_{\sigma_{\vec{x}}, \sigma_{\vec{x}'}}$ represents the bond energy between a lattice site $\vec{x}$ and one of its eight, second-order neighbours $\vec{x}'$, and $\lambda_A$ and $\lambda_L$ represent energy penalties for deviation from a designated target area ($A_\sigma$) or target length ($L_\sigma$), which we set at 50 and 5.0 respectively.

If the copy attempt does not result in a lower effective energy, the copy is accepted with Boltzmann probability:

$$P(\Delta H) = e^{-\Delta H / T}$$

(3)

where T is the "cellular temperature" of the system, a measure of cell motility, which we set at 50, and $\Delta H$ the change in the effective energy function.

In the cell elongation [14] and contact inhibition [28] models cells secrete a chemoattractant $c$, which diffuses and degrades in the ECM:

$$\frac{\partial c(\vec{x}, t)}{\partial t} = \alpha(1 - \delta_{\sigma_{\vec{x}}, 0}) - \varepsilon \delta_{\sigma_{\vec{x}}, 0} c(\vec{x}, t) + D \nabla^2 c(\vec{x}, t)$$

(4)

where $\delta_{\sigma_{\vec{x}}, 0} = 1$ outside the cells, $\alpha$ is the secretion rate and $\varepsilon$ the degradation rate of the chemoattractant and $D$ the diffusion constant of the chemoattractant.

An additional term is added to the energy function, which promotes copies upwards the chemoattractant gradient:

$$\Delta H_{chemotaxis} = -\mu(c\left(\vec{x}'\right) - c\left(\vec{x}\right))$$

(5)

Where $\mu$ is the strength of the chemotactic response, which we set at 1000. For details see [14,28].

In the mechanical cell-cell communication model [27], cells were able to deform the ECM through cell-shape dependent traction forces. The deformations of the ECM are calculated using the finite element method. An additional term is added to the energy function, such that cells respond to the ECM stiffness:

$$\Delta H_{durotaxis} = -g\left(\vec{x}, \vec{x}'\right) \lambda_{durotaxis}(h\left(E\left(\in_1\right)\right)\left(\vec{v}_1 \cdot \vec{v}_m\right)^2 + h\left(E\left(\in_2\right)\right)\left(\vec{v}_2 \cdot \vec{v}_m\right)^2$$

(6)

where $g\left(\vec{x}, \vec{x}'\right) = 1$ for cell extensions and $g\left(\vec{x}, \vec{x}'\right) = -1$ for cell retractions, $\lambda_{durotaxis}$ is a parameter which we set at 24, $h(E)$ mimics the influence of the ECM stiffness $E$ as a function of $\in_1$ and $\in_2$, the principal strains, on the tension in the cell-ECM connection, $\vec{v}_m$ gives the copy direction, and $v_1$ and $v_2$ are the strain orientations.

The preference for higher stiffness $E$ is implemented as a sigmoid function:

$$h(E) = \frac{\alpha}{1 + \exp\left(-\beta\left(E - E_{tr}\right)\right)},$$

(7)

where $\alpha$ sets the strength of the durotactic response, $E_{tr}$ determines the stiffness where half this strength is reached and $\beta$ represents the stiffness sensitivity which determines the steepness of the curve. For details see [27].

## Statistical analysis

P-values between groups were calculated using an ANOVA test followed by a student t-test. Dataset significance was defined as $p \leq 0.05$ (*); $p < 0.01$ (**); $p < 0.001$ (***); $p > 0.05$ (ns).

## Supporting information

**S1 Fig. The image analysis pipeline uses automatic segmentation to avoid bias in time-lapse analysis** A) A detailed scheme of the image analysis pipeline showing intermediate images. B) A comparison between lacunae area measured by independent peers and automatically segmented lacunae.
(PDF)

**S2 Fig. Additional images and graphs** A) ECs were labelled using membrane linker dyes and segmented from the background. B) The average area of fluorescently labelled ECs over 24 hours. (n > 600 per time step) C) The average cell length of elongated cells over time (Area between 500 and 1500 µm²; roundness below 0.25) (n > 6 per time step). D) A comparison of the number of branches per mm² for chemoattractant diffusion lengths ranging from 10 µm to 100 µm for the cell elongation model and the contact inhibition model after 2880 MCS. In magenta the *in vitro* number of branches per mm² after 24 hours. E) Cell covered area was measured as a percentage of the total well area. Shaded areas represent the standard deviation. F) ECs (white) in the mechanical model exert strain on their environment indicated by blue green heatmap. G) Histogram shows the relative frequency of lacuna areas in the computational models of endothelial cell network formation after 24 hours (2880 MCS) $\left( D = 5.0 \cdot 10^{-13} m^2 s^{-1}; \in = 1.02 \cdot 10^{-4} s^{-1}; \alpha = 1 \cdot 10^{-3} s^{-1}; n = 8 \right)$.
(PDF)

**S3 Fig. Overview images of simulated networks for different cell densities after 2880 MCS.** A) Cell elongation model. B) Contact inhibition model. C) Mechanical model.
(PDF)

**S4 Fig. Overview of dynamical analysis of simulated networks for different cell densities and diffusion length or stiffness sensitivity.** A-D) Cell elongation model. E-H) Contact inhibition model. I-L) Mechanical model. Overview images are 4000 cells after 2880 MCS.
(PDF)

**S5 Fig. Overview of branch length analysis of *in vitro* endothelial cell networks for different experimental conditions.** A) HMEC-1 cells were seeded at different densities. B) 10 mM l-lactate was added to the MCDB-131 medium prior to the start of the tube formation assay. C) HMEC-1 cells were seeded on either growth factor reduced Matrigel or regular Matrigel. HCl was added to the medium prior to the start of the tube formation assay to reduce its pH to 6.0. D) Fetal calf serum was added to the medium in different percentages. E) HMEC-1 cells were seeded on either growth factor reduced Matrigel or regular Matrigel. VEGF-A was added to the medium. F) HMEC-1 cells were seeded on either regular Matrigel diluted in DMEM high-glucose.
(PDF)

**S1 Table. Chemotaxis model parameters**
(PDF)

**S2 Table. Mechanical model parameters**
(PDF)

**S1 Video. Example video of an annotated HMEC-1 time-lapse.** Cellular networks were segmented from the background and labelled in FIJI: Branches are highlighted in cyan, lacunae in yellow, nodes in magenta and endpoints in blue.
(MP4)

**S2 Video. Example video of a CPM simulation with active cell elongation.** Simulated ECs were placed on a circular grid and run for 3000 Monte Carlo steps (MCS). ECs (red) secrete a chemoattractant (grey) that diffuses through the medium (white).
(MP4)

**S3 Video. Example video of a CPM simulation with contact-inhibited chemotaxis.** Simulated ECs were placed on a circular grid and run for 3000 Monte Carlo steps (MCS). ECs (red) secrete a chemoattractant (grey) that diffuses through the medium (white).
(MP4)

**S4 Video. Example video of a CPM simulation with mechanical cell-cell communication.** Simulated ECs were placed on a circular grid and run for 3000 Monte Carlo steps (MCS). ECs (white) exert strain on their environment (green/blue heatmap).
(MP4)

## Acknowledgments

This work was performed using the compute resources from the Academic Leiden Interdisciplinary Cluster Environment (ALICE) provided by Leiden University.

## Author contributions

**Conceptualization:** Tessa M. Vergroesen, Roeland M.H. Merks.

**Data curation:** Tessa M. Vergroesen, Roeland M.H. Merks.

**Formal analysis:** Tessa M. Vergroesen.

**Funding acquisition:** Roeland M.H. Merks.

**Investigation:** Tessa M. Vergroesen.

**Methodology:** Tessa M. Vergroesen, Roeland M.H. Merks.

**Project administration:** Roeland M.H. Merks.

**Software:** Tessa M. Vergroesen, Vincent Vermeulen, Roeland M.H. Merks.

**Supervision:** Roeland M.H. Merks.

**Visualization:** Tessa M. Vergroesen.

**Writing – original draft:** Tessa M. Vergroesen, Roeland M.H. Merks.

**Writing – review & editing:** Tessa M. Vergroesen, Roeland M.H. Merks.

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
