## [Decision Letter · Decision Letter 0]

4 Nov 2024

PCOMPBIOL-D-24-01420Falsifying computational models of angiogenesis through quantitative comparison with in vitro modelsPLOS Computational Biology Dear Dr. Merks, Thank you for submitting your manuscript to PLOS Computational Biology. After careful consideration, we feel that it has merit but does not fully meet PLOS Computational Biology's publication criteria as it currently stands. Therefore, we invite you to submit a revised version of the manuscript that addresses the points raised during the review process. Please submit your revised manuscript within 60 days Jan 04 2025 11:59PM. If you will need more time than this to complete your revisions, please reply to this message or contact the journal office at ploscompbiol@plos.org. Please include the following items when submitting your revised manuscript: * A rebuttal letter that responds to each point raised by the editor and reviewer(s). You should upload this letter as a separate file labeled 'Response to Reviewers'. This file does not need to include responses to formatting updates and technical items listed in the 'Journal Requirements' section below.* A marked-up copy of your manuscript that highlights changes made to the original version. You should upload this as a separate file labeled 'Revised Manuscript with Track Changes'.* An unmarked version of your revised paper without tracked changes. You should upload this as a separate file labeled 'Manuscript'. If you would like to make changes to your financial disclosure, competing interests statement, or data availability statement, please make these updates within the submission form at the time of resubmission. Guidelines for resubmitting your figure files are available below the reviewer comments at the end of this letter. We look forward to receiving your revised manuscript. Kind regards, Amber SmithSection EditorPLOS Computational Biology Feilim Mac GabhannEditor-in-ChiefPLOS Computational Biology Jason PapinEditor-in-ChiefPLOS Computational Biology  **Journal Requirements:** **Additional Editor Comments (if provided):****Reviewers' comments:** Reviewer's Responses to Questions

**Comments to the Authors:**

Reviewer #1: The manuscript “Falsifying computational models of angiogenesis through quantitative comparison with in vitro models” by Vergroesen et al. quantifies dynamic properties of endothelial cell networks generated in vitro and compares them with the same properties of in silico networks generated with three previously proposed computational (cellular Potts) models. Given all the different mechanisms that has been shown to generate those type of capillary networks from disperse endothelial cells, the whole idea of comparing the dynamics of in vitro and in silico vascular development, although not new, it is brilliant. It is also admirable to do own experiments, model simulations and quantification in one paper. However, there are several important points where I disagree that I hope can be clarified / reformulated before accepting the paper. I also have several suggestions to make the analysis more convincing:

- This paper deals with an in vitro assay and in silico simulations where disperse endothelial cells form a capillary network de novo, followed by a network remodelling phase. It is very confusing to call that process angiogenesis as it is widely accepted that it consist in the formation of new vessels from pre-existing ones, and obviously there are no existing vessels here. I am sure the authors know that very well, so I can’t understand their rational of using the term angiogenesis. This should be replaced throughout the manuscript by a more appropriated term such as “de novo vascular development” or “vasculogenesis”.

- The authors do not use formal parameter inference, for instance using approximate Bayesian computation like in Alamoudi et al Bioinformatics 2023. Instead, to “match” the properties of the in silico networks generated by the chemotaxis models with the ones generated in vitro, they fixed all parameters except one that is varied until observing similarities. In the mechanical model, all parameters are fixed and variations were not explored in this paper. This is a poor approach that seems like cherry picking without a convincing explanation. It is possible that there are other parameter regimes that have a better fit to the experimental situation. This is not explored in depth and should be done properly or at least clearly mentioned.

- In the experiment, there is a clear difference between the first 6 hours (aggregation, spreading and elongation) and the following hours (network remodelling). According to figure 2, none of the models seem to reproduce any of the network features in the first period, at least in the explored parameter regimes. This should be more clearly stated in the manuscript. In particular, reference 14 shows how cells exhibit significant displacement before starting to elongate. This cannot be reproduced by the elongation model where cells artificially elongate very early due to the elongation constraint.

- In the abstract, it is mentioned that the response of silico and in vitro networks are studied with respect to perturbations of key model parameters and associated experimental ones. This is a crucial aspect to investigate the validity of model mechanisms. However, in this paper this is only done for the initial cell density. Although this could be seen as a model parameter, it is not really related to the model mechanisms that are investigated, like elongation or the autocrine chemotaxis mechanisms. If the authors are not planning to investigate other key parameters, the abstract sentence mentioned above should be reformulated by a much more explicit one, mentioning the cell density variation. Also in the discussion the second paragraph should be toned down, as no “assess if systematic, concurrent parameters changes in vitro and insilico” was really done.

- The final sentence in the abstract is also overstated: “…our dynamic approach helps to clarify key endothelial cell interactions required for angiogenesis, and how the approach helps analyze what key changes in network properties can be traced back to changes in individual cell”. In fact, based on the comparison made, all three models cand be rejected. What is the mechanism in the three models that is required for angiogenesis? It is very difficult to say something about an isolated mechanisms if that mechanisms cannot be perturbed in the in vitro and in silico cases simultaneously. I think the only thing we learn is that the combination of mechanisms (and their corresponding modeling choices) present in the three analysed models are not correct.

- For the sake of reproducibility and correctness, experimental data and code should be publicly available. Publishing the data will also allow other models with different mechanisms to be compared too. It is possible that a combination of chemical and mechanical cues are both playing important roles. That was not included in any mode tested, but has been done recently with a cellular Potts model too (Chiang et al 2024 Plos Comp Bio).

Reviewer #2: Summary of the study:

This study examines three cell-based models simulating endothelial cell migration and organization during angiogenesis, assessing the validity of each model by comparing the dynamics of cell network structures obtained from in vitro experiments with those from in silico simulations. Specifically, it evaluates network similarity based on branch count and mesh area, as well as the models' responses to changes in initial cell density. The authors conclude that the model emphasizing cell elongation best captures the observed in vitro dynamics.

Major Comments:

1) Assessment of Model Similarity

1-1) The paper’s central claim rests on the similarity between the in silico models and in vitro results, with Figure 2 serving as a primary dataset. However, the basis for the assertion of similarity is insufficiently clear. For instance, in the case of lacuna area dynamics (Figure 2D), although the final output of the elongated cell model aligns most closely with in vitro data, the early-stage kinetics differ significantly. The manuscript discusses similarity by comparing only two time points, yet, as time-series data, the dynamics should be analyzed comprehensively over the entire timeline to capture the true behavior. Given the aim of this study—to identify an in silico model capable of explaining angiogenesis mechanisms—the evolution of network structure over time is a critical perspective that is currently underdeveloped.

1-2) Furthermore, in Figure 4, the response of in silico models to changes in initial cell number is analyzed only in terms of branch count. To gain a more thorough understanding, at least, the dynamics of lacuna area should also be assessed. Although the cell elongation model may show the closest resemblance to in vitro outcomes among the three models, the current data do not substantiate a strong explanatory power in an absolute sense, only in a relative comparison.

1-3) Employing more comprehensive quantitative metrics from graph theory could provide a richer, multidimensional analysis of network structure.

2) Discussion Section Speculations

In the Discussion, the explanatory power of the cell elongation model is attributed to factors such as long-range connections, slower movement along the short axis, and lower network density. Notably, the phrase “We can conclude…” in the third paragraph seems overconfident given the speculative nature of the current discussion. To strengthen these claims, the authors should perform additional computational experiments to directly test the impact of these specific parameters.

Reviewer #3: This study focuses on the study of dynamics in network formation of endothelial cells in the process of angiogenesis by comparing in vitro experimental results with the in silico computational model. In a 2D environment, the development of the EC networks was tracked by custom imaging analysis. The work highlights the strengths and limitations of the existing computational models in emulating the complexity and dynamic nature of biological angiogenesis and, therefore, provides insight into further refinement of these models.

I have only some minor comments:

The code or software should be publicly available by the authors.

While the paper describes network formation at higher cell densities both in the in vitro and in silico, it lacks the biological mechanism behind this behavior. A brief discussion on potential factors or mechanisms, such as signaling pathways or spatial constraints, would be more biologically insightful.

The limitations with 2D models and discussion about the feasibility and challenges toward shifting to 3D models would enhance the study.

**Have the authors made all data and (if applicable) computational code underlying the findings in their manuscript fully available?**

Reviewer #1: **No: ** No data or code is available. It is very important the authors provide both.

Reviewer #2: **No: ** In the "Data and Code Availability" section, I can only find the statement saying "The authors declare no competing interests."

Reviewer #3: **No: ** In data and code availability, the author just wrote no competing interests.

PLOS authors have the option to publish the peer review history of their article (what does this mean? ). If published, this will include your full peer review and any attached files.

**Do you want your identity to be public for this peer review?** For information about this choice, including consent withdrawal, please see our Privacy Policy .

Reviewer #1: No

Reviewer #2: No

Reviewer #3: No

 **Figure resubmission:**While revising your submission, please upload your figure files to the Preflight Analysis and Conversion Engine (PACE) digital diagnostic tool, https://pacev2.apexcovantage.com/. PACE helps ensure that figures meet PLOS requirements. To use PACE, you must first register as a user. Registration is free. Then, login and navigate to the UPLOAD tab, where you will find detailed instructions on how to use the tool. If you encounter any issues or have any questions when using PACE, please email PLOS at figures@plos.org. Please note that Supporting Information files do not need this step. If there are other versions of figure files still present in your submission file inventory at resubmission, please replace them with the PACE-processed versions. 
---

## [Decision Letter · Decision Letter 1]

14 Mar 2025

Dear Prof.dr. Merks,

We are pleased to inform you that your manuscript 'Falsifying computational models of endothelial cell network formation through quantitative comparison with in vitro models' has been provisionally accepted for publication in PLOS Computational Biology.

Best regards,

Amber M Smith

Section Editor

PLOS Computational Biology

Reviewer's Responses to Questions

**Comments to the Authors:**

Reviewer #1: The authors addressed all my main points and I don’t have further concerns.

Reviewer #2: Improved image analysis techniques have eliminated the previously puzzling dynamics observed during the initial response in the in vitro data. As the authors suggest, these early observations were likely affected by artifacts, and this improvement appears to have resolved the significant discrepancy between the in vitro and in silico datasets. However, to further establish the robustness and reproducibility of the improved pipeline, it would be advantageous to validate the method—such as by comparing it with established techniques or applying it to independent datasets.

In the revised manuscript, the authors acknowledge the limitation of the cell elongation model in the Discussion. Although this model effectively replicates the later remodeling phase of network formation, it assumes that cell elongation is the primary driving force—a notion that contradicts in vitro observations where endothelial cells first aggregate and spread before elongating. In light of this, I recommend reframing the manuscript to emphasize “explaining the cell remodeling process in angiogenesis” rather than the entire angiogenic process. This shift would clarify which stage of angiogenesis the cell elongation model is best suited to explain. Additionally, the manuscript should underscore the biological and clinical significance of studying the remodeling phase in both the Introduction and Discussion.

Reviewer #3: The authors have thoroughly addressed all my concerns, and I have no further questions.

**Have the authors made all data and (if applicable) computational code underlying the findings in their manuscript fully available?**

Reviewer #1: Yes

Reviewer #2: Yes

Reviewer #3: Yes

PLOS authors have the option to publish the peer review history of their article (what does this mean? ). If published, this will include your full peer review and any attached files.

**Do you want your identity to be public for this peer review?** For information about this choice, including consent withdrawal, please see our Privacy Policy .

Reviewer #1: No

Reviewer #2: No

Reviewer #3: No

---

## [Editor Report · Acceptance letter]

PCOMPBIOL-D-24-01420R1

Falsifying computational models of endothelial cell network formation through quantitative comparison with in vitro models

Dear Dr Merks,

I am pleased to inform you that your manuscript has been formally accepted for publication in PLOS Computational Biology. Your manuscript is now with our production department and you will be notified of the publication date in due course.

With kind regards,

Zsofia Freund
